# Quality Criteria to Evaluate Performance and Scope of 2030 Agenda in Metropolitan Areas: Case Study on Strategic Planning of Environmental Municipality Management

**DOI:** 10.3390/ijerph17020419

**Published:** 2020-01-08

**Authors:** María de Fátima Poza-Vilches, José Gutiérrez-Pérez, María Teresa Pozo-Llorente

**Affiliations:** Department Methods of Research, Faculty of Education, University of Granada, 10871 Granada, Spain; jguti@ugr.es (J.G.-P.); mtpozo@ugr.es (M.T.P.-L.)

**Keywords:** 2030 Agenda, strategic planning, quality criteria

## Abstract

The United Nations’ (UN) 2030 Agenda brings new governance challenges to municipal environmental planning, both in large urban centres and in metropolitan peripheries. The opportunities of the new framework of action proposed by the United Nations (UN) and its integrative, global, and transversal nature constitute advances from the previous models of municipal management based on the Local Agenda 21. This text provides evidence to apply quality criteria and validated instruments of participatory evaluation. These instruments have been built on the foundation of evaluative research, a scientific discipline that provides rigour and validity to those decisions adopted at a municipal level. A case study focused on a metropolitan area serves as a field of experimentation for this model of the modernization of environmental management structures at a local level. Details of the instruments, agents, priority decision areas, methodologies, participation processes, and quality criteria are provided, as well as an empirically validated model for participatory municipal management based on action research processes and strategic planning that favours a shared responsibility across all social groups in the decision-making process and in the development of continuous improvement activities that are committed to sustainability. Finally, a critical comparison of weaknesses and strengths is included in light of the evidence collected.

## 1. Introduction

Recent advances in the field of Sustainability Sciences open the gate to emerging disciplinary areas such as Global Urban Science [1,2,3,4], built from new paradigms and ways of doing science through models that are participatory in nature. These models provide significant novelties to the ways of developing socio-environmental knowledge and justify decision-making outcomes in municipal management. They also open an inexhaustible field of exploration for the progress and modernization of municipal governance models and urban environment management [5,6,7,8]: ‘science-policy interactions between urban scholars and urban practitioners have, in the wake of the Paris Agreement, Sendai Framework, Sustainable Development Goals (SDGs) and the United Nations’ (UN) Habitat III-New Urban Agenda (NUA), undergone important steps towards greater integration’ [9] (p. 12).

From this logic, environmental strategic planning (ESP) applied to the field of municipal management emerges as a governance instrument that provides rigour and rationality to the interventions and decisions proposed to counter environmental problems and scenarios. The principles that inspire it, and the methodologies it applies, base the actions on quality criteria endowed with instruments for evaluating achievement and compliance with standards. These instruments help to ensure decisions are made following a certain direction or to convert them following the demands and requirements of the new planning and urban governance agendas [10] such as the 2030 Agenda.

The strategic plan (2030A) launched by the United Nations (UN) in 2015 represents an integrative framework for the development of environmental governance within the framework of municipal management. This agenda inherits the spirit with which the Agendas 21 [10,11,12], which started at the Rio 92 Summit, were built, giving them continuity from a new and more comprehensive integrative framework that demands cross-cutting commitments around the Sustainable Development Goals, which contain 17 goals, 169 targets, and 241 indicators.

The 2030A takes up the torch for the advances and successes of Agendas 21, overcoming its limitations [13]. This is especially true in relation to the processes involving social intervention and its methodology; providing them with a timeline and the means and instruments of a strategic nature that go beyond the immediate, that contemplate the basic ingredients of environmental complexity, and that are approached from a holistic and inclusive vision. In addition, this vision cannot be achieved if it is not based on participatory and innovative action methodologies that evaluate management processes, consensual decisions and challenges.

Therefore, from this logic of addressing the 2030A implementation processes in municipal management, in this document, we show the steps to follow in the application of participatory methodologies for the development of 2030A in a municipal context using the strategic environmental evaluation (SEE) and participatory action research (PAR) approaches, involving researchers, citizens, and managers.

Academic environmental research can play a key role in informing the design, implementation, and evaluation of sustainable urban strategies at the global scale. In addition, the active involvement of various non-academic actors in the production of urban knowledge for policy, ‘as well as the multitude of actors involved in urban affairs (beyond government) requires the scholarly community to look beyond academia and forge new collaborations to enhance research use into urban strategies’ [9] (p. 14). Dominant research modes are not enough to guide the societal transformations necessary to achieve the 2030A. Researchers, practitioners, decision makers, funders, and civil society should work together to achieve universally accessible and mutually beneficial sustainability science [14]. New approaches to science, such as action research [15], mode two knowledge production [16], transdisciplinary research [17,18], and post-normal science [19], propose that scientists should engage in deliberative learning processes with societal actors, with a view to jointly reflect on existing development visions and create new, contextualized ones [20].

Therefore, this research aims to show how the approach to urban sustainability assessment developed from participatory action research processes can be a good methodological reference to address the design and implementation of the 2030 Agenda in contexts of local management.

Within the framework of this logic of strategic environmental evaluation addressed from participatory action research, our study starts from the following aims and research questions (Table 1).

## 2. Framework

### 2.1. The Agenda 2030 as Planning and Action Frameworks

The 2030A proposes a framework for knowledge-based transformations to sustainable development that reconciles evidence and socio-political deliberations for accelerated action [20,21]: understanding systemic interactions, understanding competing development agendas, and understanding transformations in concrete contexts.

Defining the term ‘strategy’ in the field of municipal management can be complicated because of the same complexity derived from the delimitation of the concept of management in local administration. The idea of strategic planning can be understood as the articulation of a set of operational elements aimed at establishing processes with the capacity for social and territorial transformation; processes that, in the medium or long-term, revert the conservation of ecosystems and/or in the improvement of the quality of life of citizens. In the fields of organizational management and business, every strategy involves establishing a work scheme, and designing an organized action protocol that facilitates interventions within a solid framework, and at the same time makes it possible to control external variables and factors that can influence the process, generating a competitive advantage that allows it to successfully remain in the market [22].

From this logic, we can consider municipal management as a typology of activity for municipal organizations whose priority activity is to define goals aimed at improving the quality of life and welfare of citizens in the territory they live, from approaches based on participation and democracy.

This last aspect is perhaps the main element of agreement among authors when conceptualizing the strategic elements in the field of public management, in opposition to their application within the business environment; those aspects that make mention of the idea of involvement and consensus and the need to jointly build plans that affect those involved in one way or another and that look to the future [23].

Under this action prism, the new environmental strategic planning (ESP) models require citizen participation and leadership as instruments of change and improvement. Some examples of successful case studies that have become reference models are the following: the ESP model to address issues related to the management of closed coastal seas [24,25,26]; effectiveness of ESP in Thailand to achieve the legitimacy of the processes [27]; ESP and the use of indicators in water resources planning [28]; integrated models that develop coherent and policy relevant socio-ecological strategies, which are relevant to policies where appropriate decision frameworks need to be co-developed across the range of stakeholders and decision-makers in Coastal Bangladesh [29]; integrated sustainability models in management organizations [30]; identify key governance challenges that cultivating collective action, accountability, decision spaces for stakeholder interaction regarding decision-making, investment, action, and outcomes [31]. All these approaches have the potential to integrate knowledge across the natural and social sciences, and to do so in a new way; that is, a framework to overcome ontological differences between the social and natural sciences, and thereby also overcome ontological barriers in sustainability research [24,32]. The priorities and needs are identified, defined, and planned from the consensus and unique interests of the various segments of citizenship, harmonizing demands of majorities and minorities, giving voice and a vote to all sectors of the population (from childhood, youths, the elderly, ethnic minorities, and so on). In this sense, ‘the generation and support of small foci of social change in the field of environmental sustainability seems an immense field of opportunities, which has, among other advantages, the ability to: demonstrate that another way of doing things is possible, overcome mental obstacles and prejudices about alternative solutions, normalize or improve the image of models considered exotic, if not, marginal, and, amplify the positive effects of actions that have a moderate implication’ [33] (pp. 12–13).

Traditionally, Agendas 21 for local development (L21A) have been a clear example of these small plots of social and environmental change demanded by society today, even in spite of the discrepancies, controversies, resistance, and frequent divorces that usually accompany any sphere of citizen intervention. Within the L21A processes, strategic and participatory planning acquires true meaning as a methodology of intervention and local transformation that has no reason for existence if it is not for citizen involvement and social leadership [34]. Currently, the 2030A and the Sustainable Development Goals (SDG) are the frameworks of reference guiding municipal administrative institutions, ensuring that their management model is sustainable and incorporates sustained strategic planning.

If we take as reference the 17 SDGs and 169 targets associated with them, we could affirm that the majority are linked to local competences regulated in the laws, norms, and regulations in which the municipal management is structured. This reflection highlights how transcendental the application and adaptation of 2030A is for the City Councils to comply with the SDG.

While the 17 SDGs are not legally binding treaties, there is a political and ethical commitment that must be addressed by every Municipal Program of Action for the coming years up to 2030, being a strategic priority in the achievement of local goals and thus meeting the goals of the SDG, in terms of providing basic services and promoting endogenous, inclusive, and sustainable territorial development.

This is a great challenge for City Councils at present and a pending issue. They are, therefore, responsible for the design of a strategic plan that connects their political action program with the requirements of 2030A and the SDG, taking citizenship leadership in the decision-making process as prescriptive, as well as the establishment of multi-level articulations that favour the fulfilment of all SDGs, whether or not they are municipal management competencies (Figure 1).

Along these lines, within the framework of Sustainability Science, a new emerging disciplinary field gains prominence in the form of what some authors call ‘Global Urban Science’, which plays an essential role in strategic planning processes in municipal management, as argued in the following section and whose characteristics, according to the Nature Sustainability Network, are summarized in three key messages [9] (p. 2):A new global science is needed for the urban era: there is a need to develop an ‘urban science’, not as a single science, but as a cross-cutting field of engagement across multiple disciplines;Urban science needs a broad range of experts and information: the urban science community will need to include a wide range of experts, including non-academic actors such as Non-Governmental Organizations (NGOs), residents, consultancies, industry, international organizations, city networks, and the scholarly edifice of academic research;An urbanizing planet calls upon the sciences and policymaking to rethink and enhance their relationship across complex systems: the pathways to reform and improvement of the role of science in the future of cities goes, inevitably, through multiple sectors and scales of governance.

### 2.2. Strategic Planning in Local Management

The new approaches to citizen leadership point to new-generation models of democracy based on deliberative approaches, which consists of a transition from “I” to “we” through the creation of participatory will. These models are committed to the educational value of the process in deliberative management. They emphasize the different vital stages of decision making, assess the need for debate, the exchange of arguments and the value of consensus. Shared agreements help to fulfill collective commitments, to act together and to assume each one of their share of responsibility. In the face of aggregative policies, which resolve conflicts by voting a final vote ignoring the value of consensus [35,36]. This new prism of local action brings to light the importance of framing the desirable scenarios towards which we direct the change in an explicit model that provides a base and gives ideological, political, and social legitimacy to the interventions. If these values and principles govern the intervention, ‘the contribution of citizens (participation) and the position of the rulers (leadership) become key factors in determining the reasons, foundations and interests of a strategic plan’ [37], (p. 45). In this case, these management plans become ‘Participatory Strategic Plans´; where *participation* is considered to be a tool for citizen involvement in decision-making and in the assumption of responsibilities and commitments in the construction of their future; and *leadership* is seen as the new role that governments have to assume in order to be mediators between the interests of citizens and the final decisions of those who represent those interests [35,36,38]. This movement that we advocate must overcome the citizen distrust that manifests itself in ‘anti-democratic´ and demobilization actions of civil society [39]; as disenchantment against the broken promises of democracy [40] and the disaffection that exists between citizens towards political parties [41]. Today we are witnessing the ‘emergence of new leaderships that can be associated with the expansion and reconfiguration of public space and that contribute to consolidating other representation links’ [42] (p. 35).

Therefore, in our case, by taking strategic planning to the design and implementation of sustainable municipal management models built out of the principles of deliberative democracy, we would be talking about a networked or relational municipal government model based on participation and political leadership [43,44,45]. The following diagram (Figure 2) illustrates the different poles that can result from a combination of both aspects, marking as a favourable scenario for the elaboration of strategic plans in those cases in which there is high participation and marked political leadership [46].

In general, and taking into account the four axes mentioned above, participatory strategic planning applied to the development of sustainable municipal management models is defined by four dimensions that govern its methodological process and its objectives, becoming an ideal method for the development of participatory processes at a municipal level (Figure 3).

This intervention methodology, in the case of municipal environmental management, favours the definition of future scenarios that, in this case, translates into city and territory management models adapted to demands and the starting situation, from an approach focused on consensus and citizen dialogue in decision-making.

This is the starting point of this research and the conceptual reference in the field work carried out. In municipal management, participatory decision-making must be based on management models that involve participation structures that involve all social actors: political representatives, social representatives, business, and economic references and citizens in general. The diversity of actors who have participated in the information collection strategies used in the research (farmers, managers, women, youth, and so on) and that have covered a double aim are proof of the following: diagnose local environmental management, on the one hand, and, on the other hand, to foster intervention and improvement actions generated from debate and consensus processes. The assumption of shared responsibilities among the agents involved has been one of the achievements of this participatory because the recommendations provided by these groups in the different focus group sessions were transferred to the municipal council with a series of real improvements, as will be seen later in the ‘Discussion’ section.

### 2.3. Environmental Evaluation as a Participation and Planning Tool

From the logic of the activities in strategic planning, the strategic environmental evaluation (SEE), inspired by the proposals of the PAR (participatory action research) [47], is one of the ‘most complete instruments for decision support on wide-ranging development initiatives with potential effects on the environment. The SEE has been defined as the formalized, systematic and global process for assessing the environmental impacts of a policy, plan or program, as well as its alternatives, including the preparation of a written report on the results of that evaluation and their use for the adoption of public decisions on which to account [48,49,50]. At the same time, it is considered to be a process to integrate the concept of sustainability from the highest levels at which decisions about development models are taken’ [51] (p. 27). Historically, at international level, the SEE appears in the first National Environmental Policy Act (NEPA) (EE.UU, 1969) that required reports on the environmental consequences of federal actions. In the European Union, the development of the SEE is framed in the European Council Guidelines (85/337/EEC). Its premise was to incorporate environmental assessments at all levels of ‘decision making’ [50,52].

In general, the following application process (Figure 4) can be distinguished in the SEE [50,52].

This concept, in our field of work, gives meaning to the term strategic and action research planning. The SEE intends to serve to implement a sustainable local development process that integrates evaluation and decision-making at all stages of municipal management. Without forgetting the need to monetize the options of the environment as a source of local development and respect for natural cycles, ecosystems, spaces, and species, this task stresses the role of environmental policy as an important branch, interrelated with local actions, and not as a work area that is separate to general municipal policy, so that it helps to promote an intelligent, harmonious, and sustainable development.

Evaluation plays an essential role in this environmental planning as a scientific instrument that gives quality assurances to the decisions adopted [53]. The 2030A was developed through a largely political rather than a scientific process, the goals and targets—as well as the specific indicators developed to assess progress against these goals and targets—are formulated in a limited and somewhat inconsistent way [20]. The uniqueness of the environmental planning field requires the selection of proven evaluation models, inspired by methodologies validated in practice, built on bottom-up models [54,55] in which bottom-up participation is an essential requirement in a decision paradigm that places citizens at the heart of democratic decision-making processes, from the empowerment provided by the SEE [45] and, specifically, the participatory action research (PAR) [15,56] introduced at a time when undertaking an analysis of needs and prioritizing decisions on the actions to be undertaken strategically in the short, medium, and long-term in the municipal and global context in which they develop as historical subjects.

‘Wicked’ sustainability problems, defined as problems that are multi-dimensional, appear intractable, and for which there is no one clear solution, are increasing in number and intensity [57,58]. These problems differ fundamentally from technical problems that can be isolated and controlled using standard scientific methodologies. The unique characteristics of knowledge production that can address complex sustainability problems were first defined by Gibbons and Nowtony in their formulation of ‘Mode 2’ knowledge, defined as follows: knowledge production that is applied, integrates multiple disciplines and stakeholders, is reflexive, and offers novel ways to assess quality [59].

Experiments in science–policy collaboration at the local level are fundamental. Academia and local governments should take tangible steps towards joint investments for science–policy collaboration. ‘This includes suggested practical actions such as: City-regional and metropolitan science policy mechanisms, such as urban observatories’, need to be taken seriously by both universities and local governments, but with the support of national governments and the UN system. Appoint academically-grounded ‘chief scientific advisors’ to local government to advise on evidence use in city policymaking. Include peer review processes within the production of major private sector and city network datasets, engaging in scholarly outputs as much as reports from these analyses, including clear outlines of methodologies’ [9] (p. 5).

In the case of local environmental management, the SEE will make sense as long as it is part of the decision-making process for the definition of a strategic framework for participatory and consensual intervention on the road to sustainability. The environmental assessment will be more strategic the more directly it is associated with the decision-making process. To capture and materialize the SEE, it is necessary to take into account a series of conditions that will guide the process, among which we can highlight the following [51] (Figure 5).

The SEE, therefore, is presented as a tool for participatory strategic planning in the field of local management and aims to be an instrument that favours an analysis of the impacts of planning in the territory and in the community. On the other hand, it is proposed as a work proposal to achieve the local environmental objectives that must be assumed by the local corporation as part of its management and its policy.

The SEE is part of the territorial planning processes as a strategy of impact assessment, compliance with environmental objectives, and monitoring of policies and design of recommendations to be incorporated into management policies in a cyclical and continuous manner, based on the participation of citizens in decision making, in the search for consensus, negotiation, and in the incorporation of alternatives to local political actions [51].

In short, these are some of the premises that must be taken into account to draw up a strategic plan: the need to develop rigorous evaluative research processes, implement PAR methodologies, apply change assessment instruments, promote tools for citizen participation and for the analysis of inclusive needs that involve all sectors of citizens, and mobilize municipal management actions from the bottom-up that democratize environmental decisions.

Following this descriptive situational analysis of the potential of strategic planning in local environmental management addressed using the SEE’s approaches, a case study is presented to validate the model advocated in this research, focused on a metropolitan area.

## 3. Materials and Methods

### 3.1. Methodological Framework

Structuring a participatory strategic plan and implementing it across all its phases requires a coordinated, dynamic, and flexible process that favours citizen participation and decision-making, taking as basis the search for consensus and the prioritization of needs. Community-engaged, action-oriented research approaches involve communities that are impacted by the issues being studied. Such approaches include the overlapping traditions of participatory action research and community-based participatory research [60,61]. These processes manifest as a complex framework that requires continuous adjustments about negotiation and agreements, and re-adjustments around local management based on participatory and direct methodologies.

It is a model immersed in a structure that overcomes political-administrative management and favours transversal actions of citizen leadership across each of the stages into which it is divided. Reaching the balance between political action and social action is the essential ingredient that guarantees the success of this type of methodological structures based on teamwork, the search for consensus, the adoption of responsibilities, and decision-making in the definition and launching of the strategies.

The model that we intend to validate with this action research process, following the logic of strategic planning and through adapting the model of SEE [43,62], advocates sustainable municipal management such as the proposal shown below (Figure 6).

Once the model is defined, the exemplification of each of these phases is complex. The research that we include in this article brings forward the framework used at the beginning of each of these phases, so it has been indispensable to carry out complex triangulation processes both at a level addressing the techniques and key agents and informants, which have resulted in clues, suggestions, strengths, and weaknesses, in order to ultimately establish and extrapolate the results, in order to design quality criteria that we have deemed essential to defining a quality and sustainable municipal management model based on the philosophy of participatory strategic planning linked to compliance with 2030A and SDGs.

From this logic, this model incorporates an integral system of indicators for sustainability [63] from the approaches of a participatory diagnosis backed by technical indicators (territorial framework, water, energy, or mobility, among others); social indicators (population, economy and employment, heritage, or education, among others); and sustainability indicators such as the following: (1) Citizen perception of environmental and social problems. Prioritization. (2) Citizen representation in the diagnosis of local environmental management. (3) Level of commitment and responsibility of the different social, political, and technical groups in the process of socio-environmental transformation. (4) Definition and consensual design of the proposals and lines of action. (5) The decision-making process in the participatory diagnosis of local sustainable management. (6) The communication process as a training strategy.

Attending to this multidimensional indicator system, and as an example of this complex action research process, of the implementation of a strategic planning process based on the processes of individual and community reflection that have been carried out throughout the research, we exemplify this complexity with the analysis resulting from the diagnostic phase (steps of SEE—screening/scoping/basic information) of one of the sustainability indicators addressed in the investigation (step 4 of SEE—prediction of impacts) (Figure 6): ‘Citizen perception of environmental and social problems. Prioritization’.

From the presentation of these results, we address a discussion related to the fulfilment of objectives. There is a methodological reflection on the process followed through the research that will enable us to validate the proposed model from five basic elements that respond to the objectives and questions proposed at the beginning of the paper (step 5 of SEE—report):Suitability of the information collection instruments used and of their quality.Strengths of the model.Weaknesses of the model.Contribution of the model for the development of sustainable local management.Feasibility and transfer possibilities to other contexts.

### 3.2. Instruments for Data Collection

To articulate this strategic framework, the instruments used for data collection are as follows (Table 2).

These information collection instruments have enabled a diagnosis and characterization of the municipality through the strategic environmental evaluation.

The application of these instruments has favored citizen participation and participatory decision making, which has allowed us to draw a real image of the municipality evaluated and, therefore, from the work groups created, to be able to profile in a participatory way, strategic lines for the improvement of sustainable local management.

One of the great criticisms that have been made to all previous models of environmental diagnosis (L21A, Environmental Education Strategies, National, or Local Strategic Sustainability Plans) has been ignorance or non-consideration of social aspects and fragmentation of all this information. The advantage of this model is that it provides an integrated diagnostic model that rotates in the structure from the bottom-up, as mentioned above.

### 3.3. Context, Sample and Agents

The study was carried out in a municipality in the metropolitan area of the city of Granada (South Spain, Europe), located 7 km from the capital, considered to be a “dormitory town” (linked to work in the capital), with approximately 20,000 inhabitants, and whose main economic sources are agriculture and the service sector (Figure 7).

The municipality is distributed in six population centers and is governed by a conservative political group, but one that promotes the design of policies and strategic plans for local development.

The City Council has different strategic plans, such as ‘municipal action plan’, ‘action plan for sustainable energy’; ‘first local health plan’, ‘social tourism program’; ‘economic-financial plan’; and ‘programs for equality’, for ‘youth’, among others.

On the other hand, the transparency portal, the citizen service, the consumer information office, and the youth information center, among other services, promote the development of these policies based on more participatory and bidirectional work proposals between political organizations and civil organizations.

This study generated a process of citizen reflection where all the key agents of the municipality have played a leading role as informants who have their own requirements. All citizens were involved, diversifying the sample as shown in the following tables (Table 3).

### 3.4. Analysis Procedure

The analyses carried out on the information collected were of a different nature depending on the technique used. We followed a mixed methodology to analyze quantitative and qualitative information with software for data analysis and treatment; that is, SPSS v.23 for the analysis of quantitative data, and Nudist Vivo v.10 for qualitative data.

With regard to the validity of the questionnaire, we highlight that the analysis of different documents related to the subject and other instruments used in previous studies and the consultation of a group of experts has allowed us to guarantee the validity of content; we have ensured the construct validity through a factorial analysis, and the criterial validity through the correlation of all the items of each of the blocks involved with the total of each of them (with the exception of itself), having, for the majority, obtained Pearson’s product-moment correlation coefficients, statistically significant at alpha levels of 0.01 and, to a lesser extent, at 0.05.

For the calculation of the reliability of this questionnaire, we used the internal consistency procedure. The results achieved in Cronbach’s alpha per instrument and thematic blocks are satisfactory [64], ranging from 0.70 to 0.86, as shown in the following table (Table 4).

Another indicator that supports this consideration is the presence of reliability coefficients of little or no gain, if not some loss, when we have eliminated, one-by-one and in various rounds, each of the items that made up each thematic block.

Regarding the qualitative information (discussion groups, citizen participation forums, letter to municipal political representatives, monitoring commission), we based our analysis on the four quality criteria that need to be considered in the analysis of qualitative information (credibility, applicability, consistency, and neutrality) [65,66]. (i) Credibility: during the analysis process, conversations were held with participants in the study to corroborate the interpretations made based on their answers. (ii) Applicability or transferability: the study was carried out in only one municipality of the province of Granada, but instruments and results obtained can be applied in other contexts with similar characteristics. (iii) Consistency: we consider that similar results would be obtained if the study was to be replicated in other municipalities because the analysis was carried out in a meticulous way from a process of triangulation of sources and techniques. (iv) Neutrality: the detailed description of the research process carried out indicated in this article shows that it was a neutral and non-biased process.

## 4. Results

Next, we broadly present the most relevant results achieved after an analysis of the information collected through the different instruments used in the diagnostic phase for the indicator ‘Citizen perception of environmental and social issues. Prioritization’, with the dual purpose of (1) defining the socio-environmental problems of the municipality from the citizen’s perception; and (2) establishing lines of action that allow us to justify the validation of the model presented in this article as lines of action and as a proposal for the development of sustainable management at a local level.

### 4.1. Citizen Opinion Questionnaire

To identify the socio-environmental problems from the perception of citizenship, a factorial analysis (the type of factor analysis calculated is exploratory; the extraction method used was that of main components and the rotation method, varimax with Kaiser normalization, that is, eliminating components with a percentage of explained variance under 1% (<1)) of the answers given in the questionnaire was undertaken in order to identify response patterns, or whether these are related across common dimensions.

Through the analysis, we were able to identify five factors that, together, explain 48.26% of the total variance, with a first factor that explains 10.94% of it, and the rest that range between 7.96% and 10.75% of variance explained. The values achieved by the communalities are between 0.14 and 0.67, and indicate the acceptable representation that the items included in the scale have acquired.

Finally, Bartlett’s sphericity test, with a value of 164.70 and a *p* = 0.000, and the KMO (Kaiser–Meyer–Olkin) sample adequacy measure, with a value of 0.814, allow us to state that the correlation matrix is not an identity matrix [67]. Therefore, there are a number of significant high inter-correlations, as the value found in the Bartlett test is significantly high [68]. This, together with the value obtained in the KMO test, a meritorious value [69], and the value obtained by the determinant of the correlation matrix (R = 0.016) indicates that the data matrix is suitable for the factor analysis (Table 5, Table 6 and Table 7).

### 4.2. Monitoring Commission

The monitoring commission was intended to be the participatory body that guided and evaluated the process of diagnosis and implementation of the local environmental management model. This entails the presence of a relevant social representation to ensure that most of possible perspectives are present in the process, in order to approach a vision as integral and real as possible.

The following table shows the decisions made by this participation body (Table 8 and Figure 8).

### 4.3. Discussion Groups

The four discussion groups included councillors and technicians, farmers, women, and young people. The results obtained make visible the problems detected by the participants across the different socio-environmental areas according to the importance given to each one (Table 9, Table 10, Table 11 and Table 12).

### 4.4. Citizen Participation Forum

The problems detected in the different areas and their importance were also expressed by the citizens participating in the first citizen participation forum (Table 13).

### 4.5. Letters to the Council Representatives

Children and adolescents also participated in this process through letters addressed to municipal representatives. An analysis of the 366 letters written allowed us to prioritize the needs identified by these groups, as shown in the following table (Table 14).

## 5. Discussion

Carrying out the socio-environmental diagnosis of a municipality from the citizen’s perception held by the different agents of the community requires a process of the triangulation of information, as well as synthesis and prioritization. Thus, the SWOT ((Strengths—Weaknesses—Opportunities—Threats)) analysis and the triangulation of the information collected in the diagnostic phase through the various participatory instruments used allowed us to identify the most urgent and greatest priority needs, as shown in Figure 9.

The problems addressed by the population living in this municipality through the different participation processes provided us with a generic vision of the environmental and social situations of the municipality studied, as well as its limitations, which will enable us to subsequently develop action strategies that minimize existing needs.

Within each of the areas in which the needs and problems detected in the municipalities are grouped, there are first-order problems that need to be addressed during the next stage of the implementation of this management model: the action plan.

If we take into account the principles in which 2030A is framed, making that diagnosis and addressing environmental issues in an integral way means to address them from a double perspective: on the one hand, we must analyse the objective data of the reality of the environment (physical–environmental diagnosis) and its associated problems and, on the other, understand the perception and assessment that citizens make of it (participatory diagnosis) [10,11]. From this logic, the problems derived from citizen perception and assessment linked to the SDGs are summarized into the following broad categories (Figure 10).

Finally, and in response to the objectives of the study, we can respond to them by addressing the results from a quadruple approach:(a)*Methodological reflection on the model*: 1. The design and implementation of the participatory strategies presented in this study enabled us to collect information on the citizen’s perception of the environmental and social problems existing in the municipality studied, as well as possible proposals for improvement. 2. Foster the participation of the population in decision-making related to local management in the environmental field. 3. Promote a process of personal and collective self-reflection that favours addressing any doubts and assumptions of responsibilities by the population in the process of developing a participatory local environmental management model endorsed by the principles of 2030A.

These information collection strategies enabled the PAR model because some of these participation platforms were permanently established in the municipality (i.e., monitoring commission) to make a formative evaluation of the actions and to make binding decisions regarding sustainability in the daily local management of the investigated municipality. This commission, in fact, made decisions related to issues to work in the discussion groups, and in the forum, they voted the election of the logo, established the internal regulations of the commission, and decided on the first actions to be carried out based on the report obtained in the diagnostic phase, among others. In the process of conducting a socio-environmental diagnosis for the implementation of this innovative local management model, there are lights that strengthen the process and shadows that weaken it:(b)*Strengths of the validated management model*: 1. Allows gathering of broad perceptions of the population, facilitating the participation of all citizens in the process. 2. Facilitates initial contact with the population in order to consolidate much more complex structures of citizen participation. 3. Systematizes and structures procedures for collecting information that encourage citizen participation in municipal management. 4. Provides a formula for collective and individual reflection of the citizen in relation to their behaviours and attitudes within the global municipal structure. 5. Facilitates the knowledge of the premises of the 2030A and its application by the neighborhood. 6. Favours the involvement of all those most representative in the municipality.(c)*Weaknesses*: 1. Difficulty in ensuring the representativeness of the entire population and that the demands described are really those that exist and not just a reflection of individual issues. 2. Political opportunism conceived as occasional strategies of a circumstantial nature that are intended to merely meet political and economic targets from specific subsidies. 3. Risk of becoming decontextualized and discontinuous actions that do not facilitate results in the medium or long-term. 4. Lack of motivation and trust in these types of structures by the population, including political groups and the municipal technical group. 5. Compliance with expectations.(d)*Contribution to Participatory Municipal Management and feasibility of application to other contexts*: Among the contributions of the municipal management model we supported in this study, and which can serve as a reference and be suitable for application in other contexts, we highlight the following: 1. Provides information for contextualized management. 2. Gives ground to the political and local management actions carried out, which enables a relevant degree of success and effectiveness. 3. Introduces the population to innovative processes of citizen participation and local development. 4. Consolidates reflective processes and continuous training in the development of sustainable actions. 5. Promotes the involvement of representative social sectors in the municipality in the decision-making process of municipal management.

In general, therefore, we can say that this study reflected on how the SEE can be applied in sustainable local management. In the document, the first five steps in the application of the SEE model were exemplified, as a previous step to the design and application of the actions. In addition, this model is defended from the logic of the PAR.

Some of the most outstanding strengths that we can point out in light of the results detected are as follows. The monitoring commission was established as a platform for stable participation in the municipality, which has led to the generation of public initiatives (i.e., creation of a peri-urban park, mobility plan). Short-term evaluations and evaluations based on the permeability of government teams were made to the concrete proposals derived from the participatory diagnosis made with all the agents involved. The model advocated in research is a model applied and transferable to other contexts.

There are limitations became visible with this study, such as the shortage of stable citizen participation platforms. Controversies that arise when seeking consensus between political, social and economic representatives; as well as citizen and institutional obstacles to implement the recommendations that arise from participatory processes. The scope of the research was not contemplated in the medium- and long-term impact assessment (only in the short term). All these constitute future lines and research challenges.

## 6. Conclusions

As conclusions, we can highlight that this research has contributed with key elements for the theoretical and methodological body of the SEE and the ESP, such as the following: (1) The entire process developed in this investigation has had a formative character for the different agents involved, including the municipal government team and opposition political groups. (2) All the agents involved have experienced the dynamics of participatory work in the first person and examples of information gathering instruments that support and enrich the decision-making process taking into account the diversity of interests and opinions expressed in the process. (3) The research has allowed a matrix of criteria and indicators, with which they are working for the improvement plan. These criteria and indicators focus on analyzing the SEE application process and its associated steps, taking into account the adaptation of the case study model (according to the model described in Figure 6) and arising from this process of continuous reflection that was present in all the work. It is also nurtured and based on the conclusive data and the results of this study, taking the principles of 2030A and SDG as a reference.

Taking these research conclusions as a reference and taking into account that, by 2050, 70% of the planet’s population will be concentrated in large urban centers, and by 2100, this percentage will reach 85%, great sustainability challenges are staged that involve placing cities and metropolitan areas at the heart of the issue. As has been raised throughout this work, it is a priority to respond to this problem from the decision-making process and from the leadership of the citizens. Application of the principles of 2030A represents an important challenge in addressing these challenges related to the modernization of urban management models. Decisions about human mobility, car traffic, transportation, pollution, urban planning, urban infrastructure planning, collection, treatment and waste management, lighting, tourism, water supply, garden irrigation, maintenance of green spaces, and so on require models of intelligent decision-making in which citizen participation is in the DNA of planning and management [70]. Leaning into participatory strategic action methodologies in management plans means betting on an intelligent, sustainable, and networked government model that favours the harmony between the natural and artificial, which stimulates the balance between social, environmental, economic, and political dimensions, aiming to improve the quality of life from dialogue, reflection, and citizen involvement in the decision-making of local management from a global perspective. The technological instruments at the service of the SmartCity must facilitate a creative participatory management process of decision-making that is informed, consensual, and grounded, which measures itself against taking advantage of the opportunities offered by the digitalization of a large number of processes in which the citizen can contribute and provide relevant information in real-time [71].

## Figures and Tables

**Figure 1 ijerph-17-00419-f001:**
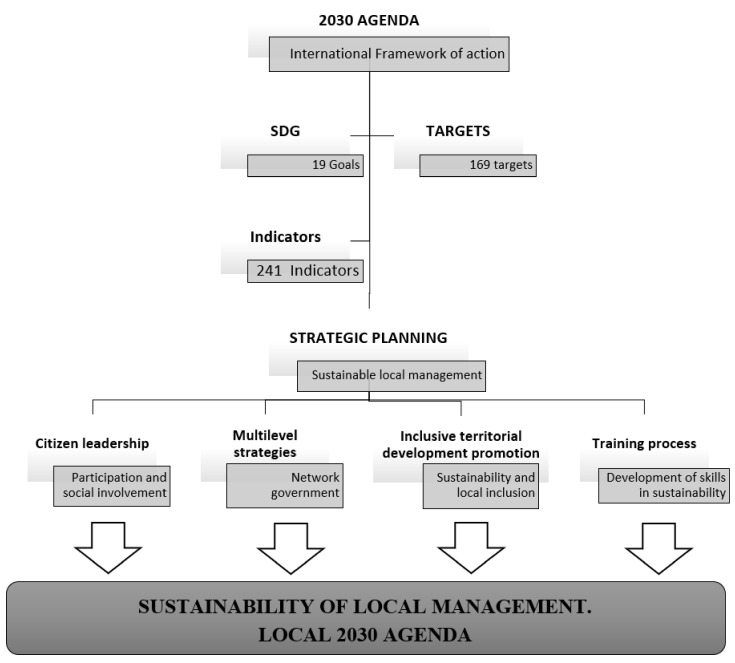
Principles of sustainable local management for a Local 2030 Agenda (L2030A). SDG, Sustainable Development Goal.

**Figure 2 ijerph-17-00419-f002:**
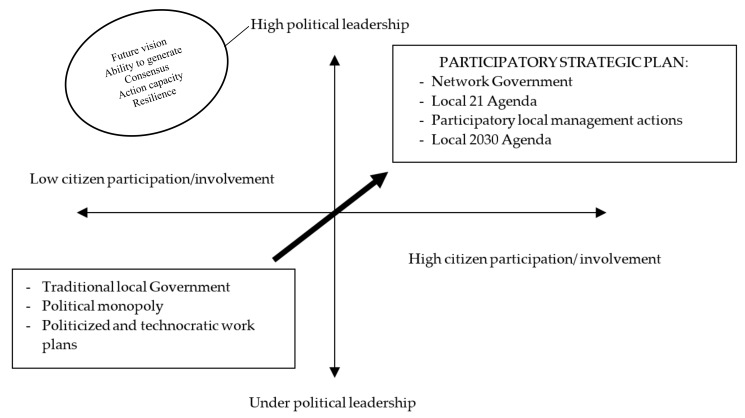
Network Government (authors’ elaboration from the work of Font, 2001 [46]).

**Figure 3 ijerph-17-00419-f003:**
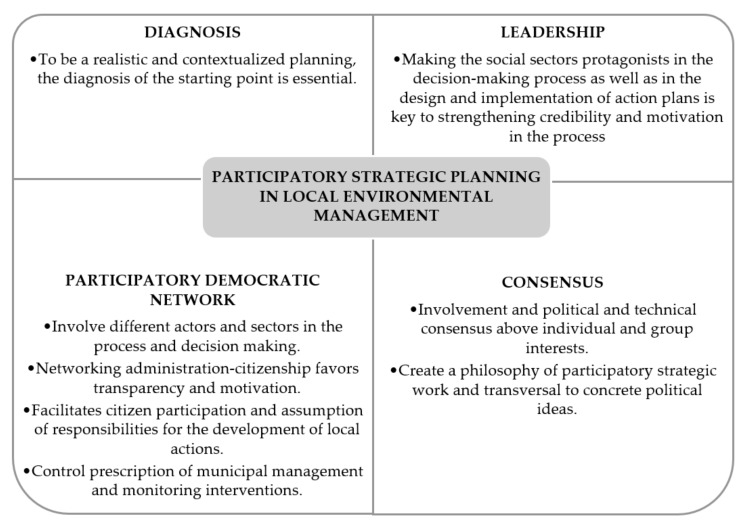
Participatory strategic planning in municipal environmental management.

**Figure 4 ijerph-17-00419-f004:**
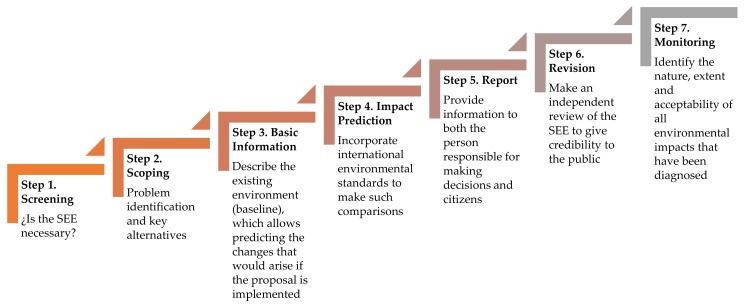
Strategic environmental evaluation (SEE) process.

**Figure 5 ijerph-17-00419-f005:**
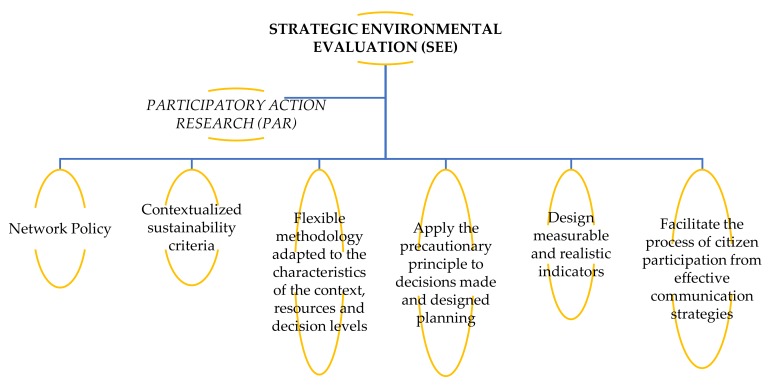
Strategic environmental evaluation: principles of action.

**Figure 6 ijerph-17-00419-f006:**
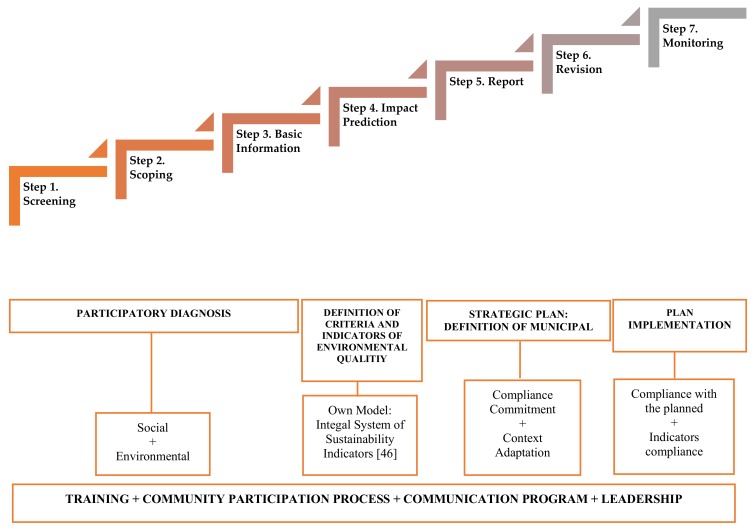
SEE model for participatory municipal management applied in our case study [63].

**Figure 7 ijerph-17-00419-f007:**
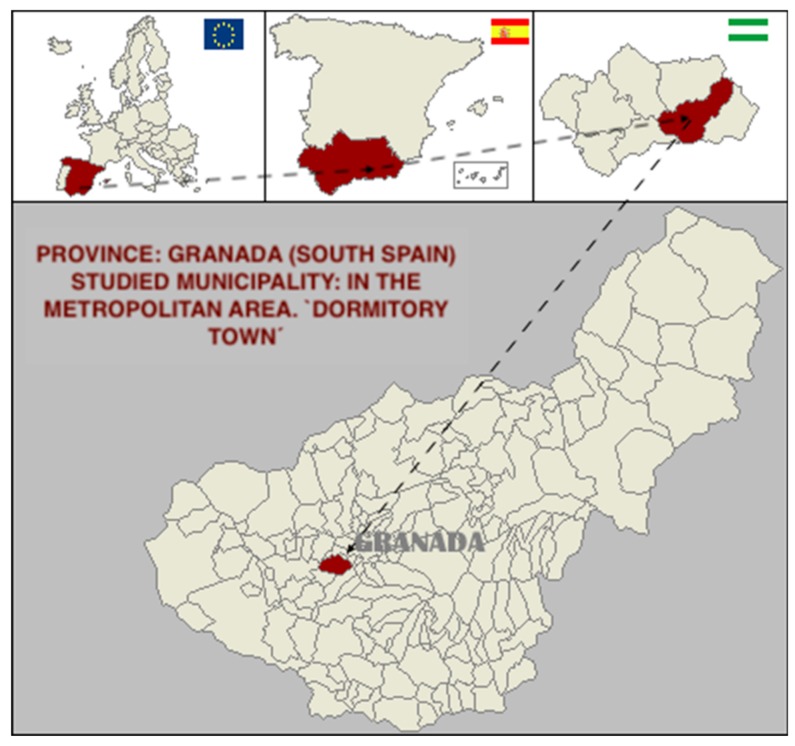
Territorial map of the municipality studied.

**Figure 8 ijerph-17-00419-f008:**
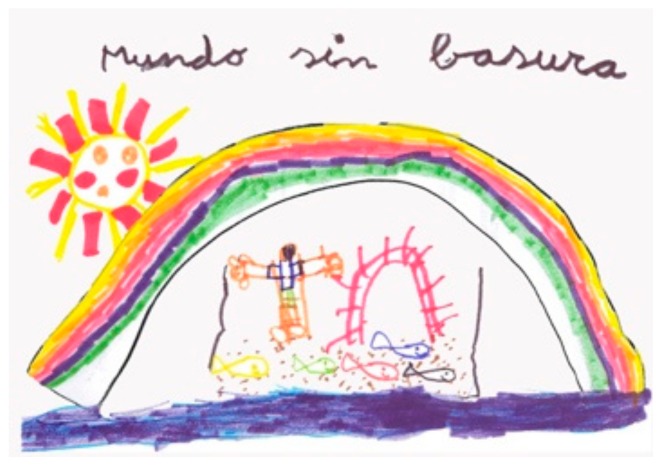
Logo referring to the sustainable local management model of the municipality. Approved by the monitoring commission and designed by ten-year-old children, “world without rubbish”.

**Figure 9 ijerph-17-00419-f009:**
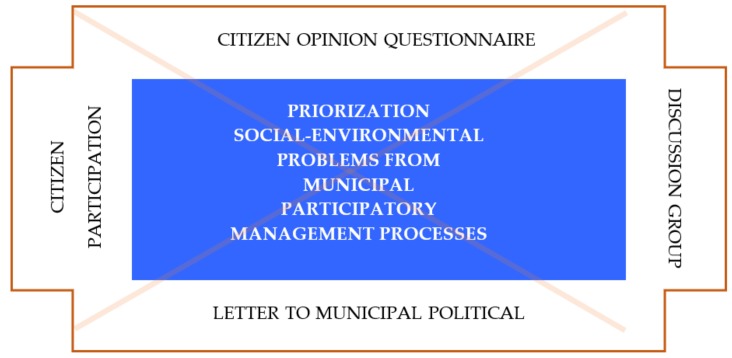
Triangulation process of environmental problems.

**Figure 10 ijerph-17-00419-f010:**
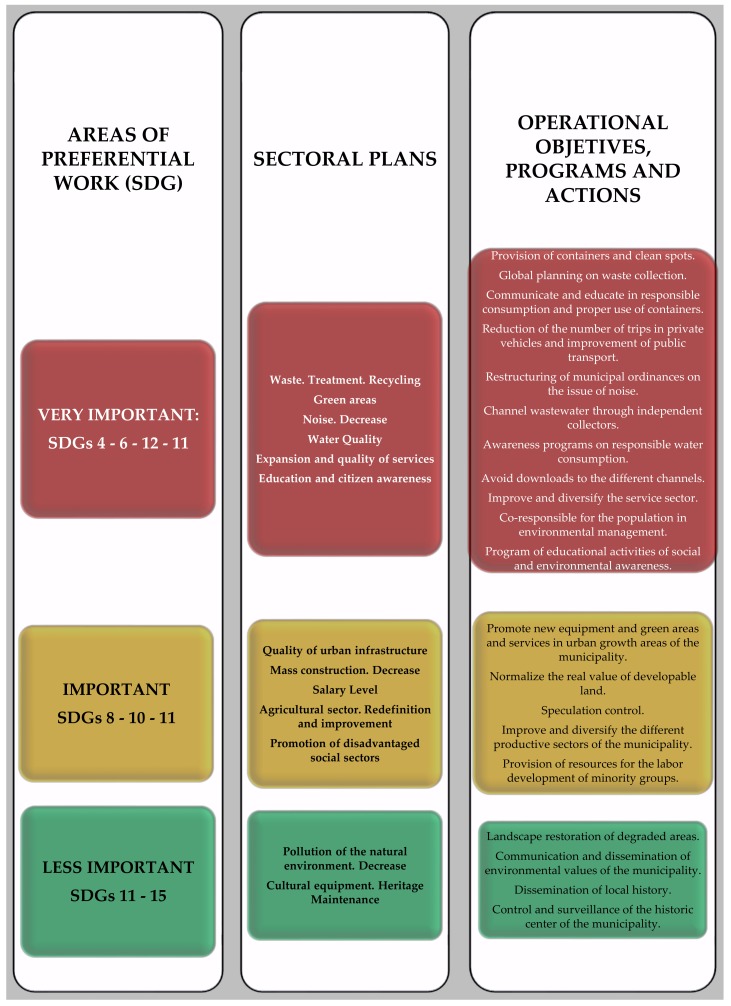
Perceived problems and SDGs. Triangulation (adaptation matrixed participatory process of Garcia-Ayllon, S, 2018 [24]).

**Table 1 ijerph-17-00419-t001:** Aims and research questions. SDGs, Sustainable Development Goals; PAR, participatory action research; SEE, strategic environmental evaluation.

Aims of Research	Research Questions (RQs)
Characterize a methodological model of strategic environmental planning based on democratic evaluation (participatory research action approach): define stages, obstacles, conditions, and limitations from a practical case study. Analyse and assess the contribution of this strategic planning model to the development of the 2030A in the case study analysed. Provide the necessary guidelines to address the 2030A in local municipal management through a citizen leadership model. Identify new challenges set by the 2030A for the strategic and sustainable management of municipalities. Define and model the planning and management stages, and analyse the possibilities of transferring these to different contexts.	RQ1. What are the novelties that the 2030A framework brings to sustainable municipal management? RQ2. What stages do the new methodologies associated with collaborative, transdisciplinary, and action research models involve for the grounds of municipal decision-making? RQ3. What criteria and quality indicators should be required from processes and instruments? RQ4. What are the most significant weaknesses and strengths of this new stage of municipal planning and management? RQ5. What viability and transfer possibilities do these new management models have in implementing them in different contexts?
**Aims of Case Study**	**Research Questions of Case Study (CS)**
Characterize the socio-environmental situation of the municipality studied from the SDGs. Identify and prioritize the social and environmental needs that arise in this municipality according to the SDGs. Collect the opinions and perceptions of citizens regarding environmental issues. Involve all social sectors and the population in general, in processes of participation and decision-making in municipal environmental management. Channel communication and dissemination processes of the 2030A and SDGs. Promote communication, participation, negotiation, and reflection processes to prioritize collective needs. Define by consensus on environmental quality indicators for the implementation of 2030A and SDGs. Define by consensus some lines of action that favor the improvement of the socio-environmental situation of the municipality under investigation. Create stable citizen participation structures that facilitate municipal decision making.	RQ(CS)1. Taking as reference the SDGs, what image of the studied municipality is projected? RQ(CS)2. What participation and communication platforms have been generated as a result of implementing the PAR? RQ(CS)3. After the SEE is carried out in the municipality, what responsibilities do the agents involved in the decision making assume? RQ(CS)4. What quality criteria and indicators have been generated after the application of the SEE? RQ(CS)5. What lines of action have been generated following the participatory diagnosis carried out?

**Table 2 ijerph-17-00419-t002:** Information collection strategies.

Strategy	Objectives	Description	Use (Phase)
Citizen opinion questionnaire	- Opinions and perceptions of citizens in environmental matters. - Involve the population in municipal participation processes. - Communicate the participatory environmental management process. - Promote that this local management model is known by all citizens.	Section 1: Environmental situation of the municipality. Item 1. What do you think are the most important environmental problems in your municipality? Item 2. Which of the following things do the neighbors of your town and which do you usually do? Item 3. What is the current situation of different aspects of your municipality? Section 2: Local environmental management in the municipality Item 4. What degree of responsibility do the different social groups have in the protection of the environment and in the socioeconomic improvement of the municipality? Item 5. You indicate the order of importance of the following solutions linked to the improvement of the environmental management of your municipality. Item 6. What do you think will improve the local sustainable development agenda in your town? Item 7. Which sector should be further developed with the realization of the local sustainable development agenda for your people to improve? Item 8. How far would you be willing to get involved so that improvement actions were carried out? Item criterion: Municipal environmental situation (in general)	- For all population sectors from 12 years old. - Diagnostic phase: To correctly direct the actions and strategies of action and participation.
Monitoring commission	- Promote a process of collective reflection from the monitoring of actions. - Consolidate a platform for monitoring and evaluation of the local environmental management process developed. - Involve the population in decision-making processes in local management.	A control and evaluation body of the local environmental management process has been created. There have been different control sessions to ensure compliance with the actions. Representatives of the different social groups in the municipalities studied participate in this commission. In this commission, issues relate to the following: - functions of the commission; - rights and duties of the commission; - election of commission members; - monthly commitments for the sustainable management of the municipality.	It has been carried out during the diagnostic phase owing to the relevance of this body throughout the process.
Discussion groups/citizen participation forums	- Promote a process of collective reflection. - Establish socio-environmental and participatory management indicators. - Triangulate the information collected with the different techniques. - Involve the population in decision-making processes in local management.	- Two citizen participation forums - Four groups discussion with two sessions: (1) councillors and technicians; (2) women; 3 (farmers); (4) youth Forum 1/Session 1 Discussion group: Characterization of the town. Prioritization of the problems.Forum 2/Session 2 Discussion group: Reflection and negotiation of intervention strategies.	At the end of the diagnostic phase and beginning of phase 2 (design of indicators): citizens contribute to the consensual definition of indicators and action strategies from the results obtained in this initial diagnosis.
Letter to municipal political representatives	- Extract the perceptions and opinions of the child population about the environmental problems of their municipality. - Promote the participation of the child population in the development of this model of participatory local management. - Raise awareness among the youngest population in care and respect for the environment. - Involve the education system in municipal management processes.	Taking advantage of the Christmas season, an activity has been developed, with the elementary courses, entitled “Letter to municipal representatives” to reflect the situation of the municipality from the point of view of children. 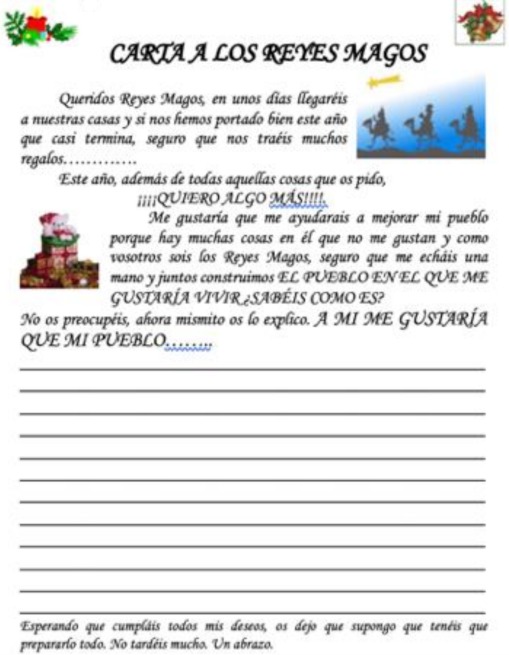	- Diagnostic phase: strategy linked to the participation plan, and the communication plan. Aimed at children.
SWOT TECHNIQUE‰(Strengths—Weaknesses—Opportunities—Threats)	- Agree and negotiate problems and solutions. - Favour a process of collective reflection. - Triangulate the information collected with the different techniques and according to different population sectors. - Promote the establishment of socio-environmental indicators. - Reference for the action plan.	First phase: problems are reorganized into weaknesses, threats, strengths, and opportunities considering the internal level and external elements. Second phase: the data are crossed and the proposals and action strategies are elaborated. Immediate actions are prioritized and established.	Transversal action: diagnostic phase, criteria and indicators design phase, and action plan design phase: the research team with a heterogeneous work group formed by process participants identifies these elements to implement the action plan.

**Table 3 ijerph-17-00419-t003:** Sample Total (N = 966).

Citizen Opinion Questionnaire (Sub-Total N = 507)										
Study level	No studies		5		Current activity			Student		424
	Primary studies		51					Employee		54
	Secondary studies		432					Unemployed		14
	University studies		19					Others		15
Age	Less than 15		193		Professional activity			Student		421
	15–25		239					Services and culture		25
	26–35		24					Housewife		24
	36–45		34					Industry		14
	46–55		10					Retired		7
	56–65		3					Official		9
	More than 65		4					Others		5
Gender	Women		257					Non-official executives		2
	Men		250							
**Monitoring Commission (Sub-Total N = 14)**										
1 representative of farmers and livestock.						1 representative of the women’s associations.				
1 representative of sports and cultural associations.						1 representative of shopkeeper associations.				
1 representative of youths						1 representative of retirees and pensioners.				
1 representative of the neighbourhood associations.						1 representative of the parents’ associations.				
1 representative of environmental associations.						1 representative of teacher of the educational centres.				
1 repres. with recognized prestige in environment/university or research institute. 3 repres. of the political groups with representation in the town council										
**Discussion Groups (Subtotal N = 30)**										
Concillors: 4		Technicians: 5		Farmers: 6			Women Association: 6		Youth: 9	
**Forums 1 & 2 (Subtotal N = 49)**										
Forum 1 (N = 23)						Forum 2 (N = 26)				
- Political groups women group: housewives and working women. - School community: teachers. - Representatives of associations. - Representatives “media”: radio and photography. - Administration technicians: sociocultural animator, woman informant. - Environmental volunteer representatives. - Youth group.										
**Letter to Municipal Political Representatives (Primary education students) (Subtotal N = 366)**										
**TOTAL SAMPLE N = 966**										

**Table 4 ijerph-17-00419-t004:** Analysis of the reliability of the citizen opinion questionnaire.

Instrument	Value α de Cronbach	Elements/Subitems	Sections of Items
Citizen opinion questionnaire	0.79	21	Environmental problems (item 1)
	0.73	22	Things that the residents of the town do (item 2)
	0.77	8	Responsibility of social groups (item 3)
	0.88	24	Current situation of the municipality (item 4)
	0.80	9	What can improve participatory local environmental management in your town (item 7)
	0.70	8	Sector to be developed with this management model (item 8)
	0.70	5	Global claims (item 10)

**Table 5 ijerph-17-00419-t005:** Kaiser–Meyer–Olkin (KMO) and Barlett test.

KMO and Barlett Test (Question:Environment Problems)		
Kaiser–Meyer–Olkin sample adequacy measure		0.814
Bartlett’s sphericity test	Chi-square approximate	164.70
	gl	210
	Sig.	0.00

**Table 6 ijerph-17-00419-t006:** Λ total and % variance.

Λ Total	2.29	2.25	2.02	1.87	1.67	48.24%
% variance	10.94	10.75	9.64	8.94	7.96	
% accumulated	10.94	21.69	31.33	40.28	48.24	

**Table 7 ijerph-17-00419-t007:** Extraction method: analysis of main components. Rotation method: varimax normalization with Kaiser.

Matrix of Rotated Components (a) “Environmental Issues”	Components					
	1	2	3	4	5	Communalities
**FACTOR 1. Environmental-context problem**						
Lack of care and cleanliness of the environment	0.73					0.61
Pollution rivers and vegetation and forest areas	0.79					0.67
Loss of landscape and agricultural land	0.44					0.38
Discharge of illegal taste on the outskirts of the municipality	0.63					0.45
**FACTOR 2. Labour problem**						
Lack of stable work		0.77				0.61
Jobs that require low training and qualification		0.57				0.43
Low salaries		0.71				0.57
High number of unemployed		0.69				0.61
**FACTOR 3. Executive-legislative problem**						
Lack of communication between municipal political representatives: put political interests before social needs			0.63			0.60
Poor coordination between town council technicians			0.67			0.59
Urban growth			0.54			0.37
Lack of urban planning			0.63			0.52
**FACTOR 4. Normative-educational problem**						
Lack of green areas				0.32		0.23
Lack of awareness towards environmental problems				0.73		0.56
Lack of constant training that makes people care for and respect their environment				0.72		0.56
Weak legislation in environment that allows the guilty get through ‘in good shape’				0.51		0.36
**FACTOR 5. Technical-environmental problem**						
Recycling waste					0.55	0.39
Existence of very loud and annoying noises					0.64	0.54
The passage of so many vehicles through the town center					0.58	0.41
Lack of bins and containers					0.44	0.43
Misuse of containers and bins					0.26	0.14

**Table 8 ijerph-17-00419-t008:** Decisions achieved by the monitoring commission.

Strategy	Decisions Achieved
Creation of a newsletter	Writing and approval of the relevant contents about the municipality in environmental, social, and economic issues
Logo design that identifies this local management model	Drawing competition proposed by the commission and addressed to all elementary students (first and second year of primary school) Approval and definition of the final logo
Website	Approval of the contents to be disseminated on the website Design: technical team of the local corporation The page offers the possibility for the population to participate through forums and virtual surveys on social, economic, and environmental issues and to know the actions that are being carried out in local management

**Table 9 ijerph-17-00419-t009:** Socio-environmental problems extracted from the discussion group ‘councillors and technicians’.

Discussion Group: Councillors and Technicians		
Importance Level	Area	Problems
Very important	Recycling and selective collection of rubbish. Container use	‘Lack of containers. Selective collection’ ‘Uncontrolled focus of all types of waste, rubbish and all types of packaging’ ‘Lack of citizen awareness in the generation of waste and deposit. Respect for the collection of equipment and debris’ ‘Lack of network of clean points’
	Optimization and expansion of green areas	‘Lack of green areas and parks’ ‘Low maintenance of green areas’
	Noise	‘Noisy’ ‘Urban centre loaded with vehicles and, consequently, with smoke’ ‘Noise pollution in the urban centre’
	Water Quality	‘Poor water quality’ ‘Existence of sanitation discharges to irrigation ditches’ ‘Poor citizen awareness in the use of water’ ‘Absence of wastewater treatment plant’
	Citizen awareness	‘Lack of citizen awareness in environmental matters’ ‘Respect for street furniture’
Important	Development of the women’ sector	‘Lack of work initiatives for women’ ‘Shortage of resources that favour the insertion of women into the job market’.
	Town planning	‘Uncontrolled housing growth’
	Space adaptation	‘Few public parking’ ‘Existence of architectural barriers’ ‘Existence of industry within the urban area’
	Citizen security	‘Unsafe entrance to schools. Matching vehicles and pedestrians’
	Cleaning	‘Unclean streets and public areas’ ‘Lack of citizen awareness and respect for the cleanliness of the town’
	Population density	‘High Population density in the urban area’
	Sector involvement	‘Difficulty in developing actions where all sectors are involved’
	Training and employment	‘Few resources for training and employment’
	Health	‘Health Services Deficiency’

**Table 10 ijerph-17-00419-t010:** Socio-environmental problems extracted from the discussion group ‘farmers’.

Discussion Group: Farmers		
Importance Level	Area	Problems
Very important	Costs	‘High labour cost with respect to the product price’ ‘Land cost’ ‘Products price’ ‘High cost of phytosanitary products’ ‘Renewal of planting products (monocultures)’ ‘Expensive labour in relation to the price for which the collected product is sold’
	Water	‘Irrigation, wastewater’ ‘Wastewater’ ‘Channeling of ditches, roads’ ‘Water of the swamp’ ‘Old ditches’ ‘In winter there is plenty of water with rain and in summer it is missing’
	The product	‘Low value of corn at this time’ ‘There are no alternative fruits for this type of agricultural land’ ‘Low tobacco prices’ ‘Regarding the cultivation of olive trees, it is difficult because there is dry land’
	The job	‘Aging of the sector’ ‘Renewal difficulty’ ‘Delay in machinery in general (methods, machines, systems...)’
	Administration support	‘Support for farmers with tobacco companies’ ‘More support for cooperatives to expedite subsidies. High administrative requirements’
Important	External variables	‘The brick factories that harm smoke and dust’ ‘Ways of the valley in very bad conditions’

**Table 11 ijerph-17-00419-t011:** Socio-environmental problems extracted from the discussion group ‘women’.

Discussion Group: Women		
Importance Level	Area	Problems
Very important	Citizen awareness	‘Awareness’ ‘Respect/Education’ ‘Indifference of people’ ‘Lack of citizen collaboration’ ‘Lack of mutual respect between groups’
	Citizen security	‘Surveillance service’ ‘We cannot walk quietly at certain times through the streets’
	Development of the women’s sector	‘Municipal nursery’
	Recycling and selective collection of rubbish. Containers use	‘Container service’ ‘There is no good waste collection plan’
Important	Optimization and expansion of green areas	‘Park care’
	Town planning	‘Street arrangement’
	Cleaning	‘You cannot walk the street or square without fear of cuts and infections’ ‘Bad smells, infections, poor vision of the town’
	Economy	‘Limited family and economic well-being’ ‘Lowering of the general economy’

**Table 12 ijerph-17-00419-t012:** Socio-environmental problems extracted from the discussion group ‘youth’.

Discussion Group: Youth		
Importance Level	Area	Problems
Very important	Services	‘Lack of leisure equipment such as swimming pools’ ‘Public transport and traffic deficit’ ‘Social and community services deficit’
	Citizen security	‘Insecurity’ ‘That the mayor listen to the young people, the local police are not in the places where this insecurity is suffered’
	Culture and education	‘A space of cultural encounter’ ‘Promotion of cultural in general, in the town’ ‘Promotion of the culture of the town abroad’ ‘More resources for the library’ ‘Specific activities for young people’ ‘Training in topics such as indiscipline and classroom conflicts’
Important	Optimization and expansion of green areas	‘Lack of green areas’
	Water quality	‘Poor water quality’

**Table 13 ijerph-17-00419-t013:** Socio-environmental problems extracted from the citizen participation forum.

Citizen Participation Forum		
Importance Level	Area	Problems
Very important	Recycling: use of bins	- Lack of containers - Recycling - Separate rubbish collection plan - Use of bins - Citizen awareness
	Existence of noise	- Motor vehicle noise - Acoustic pollution - Atmospheric pollution
Important	Lack of green areas and natural environment	- Increase of green areas - Loss of natural spaces - Protection of plant species - Improvement and conditioning of existing gardens and green areas - Citizen awareness
	Education and environmental awareness	- Civic education - Environmental education for different population sectors - Lack of citizen awareness - Disrespect for the environment
Less important	Care, cleanliness, and respect for the environment	- Cleaning the environment - Street arrangement - Lack of sanitation - Dirt, aesthetic conservation of the municipality - Citizen awareness for the respect and care of the environment
	Waste	- Uncontrolled landfills
	Others	- Poor water quality - Residual collectors - Lack of public spaces - Stock of electric towers in the urban area

**Table 14 ijerph-17-00419-t014:** Socio-environmental problems extracted from the letter to municipality political representatives.

Letter to Municipal Political Representatives				
Importance Level	Area	Problems	No. Passages	%
Very important	Services	(A) Public Services	577	40%
		1. Infrastructures and equipment	443	
		2. Quality and improvement of services	114	
		3. Social services	12	
		4. Citizen security	8	
		(B) Private Services	169	
		TOTAL	746	
	Leisure	(A) Equipment	374	21%
		(B) Activities	16	
		TOTAL	390	
Important	Environment	(A) Pollution and cleaning	124	13%
		(B) Recycling	76	
		(C) Traffic	39	
		(D) Water	17	
		TOTAL	256	
	Town planning	(A) Job	179	12%
		(B) Living place	57	
		TOTAL	236	
	Civic education	(A) Pro-social behaviors	67	8%
		(B) Pro-environmental behaviors	49	
		(C) Pro-social attitudes	42	
		TOTAL	158	
Less important	Natural environment and green areas	TOTAL	102	5%
	Employment and job stability	TOTAL	8	0.8%
	Cultural heritage	TOTAL	7	0.2%

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
