# Peer review of "Quality Criteria to Evaluate Performance and Scope of 2030 Agenda in Metropolitan Areas: Case Study on Strategic Planning of Environmental Municipality Management"

_ijerph, 2020, doi:10.3390/ijerph17020419_

Round 1

Reviewer 1 Report

The article is based on an interesting and important case, but it needs to be more critical and reflective. My main comments are:

The title: Although “performant” is an actual (although very rare) word, it doesn’t fit in the sentence, it should be changed to “performance”.

Literature review section (pages 5-8): There have been many attempts at participatory environmental evaluation and environmental planning, and it is important to examine and discuss this literature, as this is the body of work that your case will be contributing to. A particularly important issue to discuss within this is participation – many scholars are critical of state-led corporatist forms of participation, which are seen as a way of “demobilizing” civil society.  

Context: The key to case study research is locating the case within its broader context. At the moment there is only one sentence (on page 11) on the broader context. It is important for the reader to have a better sense of the institutional structure of the municipality, the challenges it faces, political dynamics, the state of civil society in the region, etc.

Table 16 on page 18: There is a sentence in Spanish that needs to be taken out (“Protección de especies vegetale”).

Critical reflection: It is important to reflect on the weaknesses and limitations of the case more – what didn’t work as well as it should have, what didn’t work at all, what should have been done but wasn’t, what was the actual impact of the process, etc. The ultimate test of participatory processes is whether the processes resulted in real change, but there are often institutional challenges to taking up the recommendations that emerge from participatory processes.

Conclusion:  In the conclusion you should reflect how on how your particular case adds to the existing body of work on participatory environmental evaluation and environmental planning.

Author Response

Response to Reviewer 1 Comments

First, we would like to thank the reviewer's comments because these comments have improved the understanding of the article. In this regard, below we specify the changes made in response to these comments.

Point 1. The title: Although “performant” is an actual (although very rare) word, it doesn’t fit in the sentence, it should be changed to “performance”.

Response 1: The word “performant” has been changed to “performance”.

Point 2. Literature review section (pages 5-8): There have been many attempts at participatory environmental evaluation and environmental planning, and it is important to examine and discuss this literature, as this is the body of work that your case will be contributing to. A particularly important issue to discuss within this is participation – many scholars are critical of state-led corporatist forms of participation, which are seen as a way of “demobilizing” civil society.  

Response 2: We are aware of the availability of scientific evidence related to the use of PAR as a form of rhetoric to legitimize political decisions. In our case, this investigation arises as a demand of the context agreed by the different agents involved, not only for the political interest and as a consequence, the diversity of agents involved and the ideas expressed.

A new paragraph has been incorporated indicating the idea expressed by the reviewer (lines 186-191).

Several references of literature on have been added. They analyze the advantages of bottom-up participation models over top-down. The analysis is specified, in our research, in examples of special attention to sectors such as childhood, women and third age. We highlight the role of groups of politicians opposed to municipal government and the value of the youth of many of the representatives (lines 213-224).

On the other hand, the introduction and the framework have been restructured at a general level in response to the comments and contributions of the reviewer to clarify concepts, expand the bibliography and structure the discourse (pages 1-9).

Point 3. Context: The key to case study research is locating the case within its broader context. At the moment there is only one sentence (on page 11) on the broader context. It is important for the reader to have a better sense of the institutional structure of the municipality, the challenges it faces, political dynamics, the state of civil society in the region, etc.

Response 3: The context has been defined in more detail in section 3.3. The territorial framework of the municipality studied, political framework and priority lines of action have been incorporated (lines 405- 426).

A territorial map of the municipality studied has been included in pag. 13.

Point 4. Table 16 on page 18: There is a sentence in Spanish that needs to be taken out (“Protección de especies vegetale”).

Response 4: This sentence has been eliminated.

Point 5. Critical reflection: It is important to reflect on the weaknesses and limitations of the case more – what didn’t work as well as it should have, what didn’t work at all, what should have been done but wasn’t, what was the actual impact of the process, etc. The ultimate test of participatory processes is whether the processes resulted in real change, but there are often institutional challenges to taking up the recommendations that emerge from participatory processes.

Response 5. A final reflection has been incorporated in the "Results" section on the potential and limitations of the research carried out based on the achievements (lines 606-623)

Point 6. Conclusion:  In the conclusion you should reflect how on how your particular case adds to the existing body of work on participatory environmental evaluation and environmental planning.

Response 6. The conclusions have been restructured by eliminating table 19 that caused ambiguity. The section has been addressed, exposing the contributions that research has made to the theoretical and methodological field of SEE and ESP (625-648).

Reviewer 2 Report

Review of

 Quality Criteria to Evaluate Performant and Scope of 2 2030 Agenda in Metropolitan Areas: Case Study on 3 Strategic Planning of Environmental Municipality 4 Management

Thank you for the opportunity to review the Quality Criteria of 2030 Agenda in Metropolitan Areas. Although I do not work in the environmental municipality field, I was interested to learn about the presentation of quality criteria and participatory evaluation indices and instruments in this arena. 

The introduction unfortunately starts off too condensed which ends up being confusing. The first paragraph begins by naming many different frameworks, stating each are important, yet the difficulty for me is in getting a clear picture of what each represents and learning how they are inter-related or build from each other. For example, suddenly there is a list of 17 objectives, 169 goals, (which really should be 17 goals, 169 targets if they are the SDGs) but I have no idea where these are from, or what their intent is (though I assumed they were the Sustainable Development Goals?).  Similarly, participatory action research is named, but there is no definition presented or explanation of why it’s a better approach.  Again, much is implied and condensed, and better connections could be made. For example, there is a call for multiple stakeholders to be involved, which is part of what PAR is, but the authors don’t connect these two ideas as coming from the same orientation.

I continue to be confused in the framework section, but then some paragraphs begin to offer explanation or a place to orient the reader ie, the paragraphs below:

“From this logic, we can consider municipal management as a typology of activity for municipal organizations whose priority activity is to define goals aimed at improving the quality of life and welfare of citizens in the territory they live, from approaches based on participation and democracy.”

“Currently, the 2030A and the Sustainable Development Goals (SDG) are the frameworks of reference guiding municipal administrative institutions, ensuring that their management model is a sustainable and incorporates sustained strategic planning.”   

Each of these could be expanded to introduce this article to provide a better starting point.. The research questions and aims provide a better grounding to the article than the first paragraphs do. Yet, these research questions presented above don’t relate sufficiently to the research questions of the case study? In fact, I never get a listing of the research questions for the case study.

Figure one is very helpful. I would also say Figure two is helpful, but there is no explanation of what high and under political leadership is. So to make this figure useful, please provide a good definition of each. 

Figure three is also useful, yet I would value more discussion of why consensus decision-making is favored. I’m not sure I understand how this would be realized at a municipal level, nor is it clear who are the people involved in consensus decision-making.. (policy-makers plus citizen activists? Elected policy-makers? Managers? ) Again a lot is assumed without enough explanation.

Presentation of the SEE is an essential component of this article, but again there is insufficient information for me to grasp how it can be used, or even how it was developed in the first place. The different components are not parallel, as some are content areas and some are processes, which I found confusing.   

I would value a more straightforward approach, demonstrating how the SEE was applied, instead of another strategic planning model, figure 5.

There is a promise of elaboration and critique of the “model,” but I’m not sure at this point which model is being referred to.

Suitability of the information collection instruments used and of their quality. Strengths of the model. Weaknesses of the model. Contribution of the model for the development of sustainable local management. Feasibility and transfer possibilities to other contexts.

The data collection gets more concrete and I appreciate the specificity of who is involved, etc. And especially love the role of the “letter to the municipality” from the children, but I remain unclear what are the questions being asked, and how do these various data collection strategies relate to the SEE. 

In the results section introduction, we learn then that an indicator is being used as one of the questions for evaluation, but why didn’t I know this earlier? And where did this indicator come from?  Here is the paragraph, but does this indicator relate to a core research question: “Next, we broadly present the most relevant results achieved after an analysis of the information collected through the different instruments used in the diagnostic phase for the indicator `Citizen perception of environmental and social issues.’ ”

I’m glad to see the factor analysis of the citizen questionnaire, but the factors should be named when the % of variance is presented. Right now I’m left to wonder what the content is until I get to the table.

On the whole, this article presents several participatory methodologies that I find to be intriguing built on democratic participatory values for municipal planning with homage to various international documents and frameworks, but I find the whole presentation confusing.  Part of the challenge is the translation to English from Spanish, and the different ways the two languages present arguments. Part of the challenge however is that the story is not presented in a straightforward way. It’s a valuable story and I would encourage the authors to make revisions.

I have several major critiques.

Use of models and frameworks. Although many are mentioned, with images shown, there is insufficient information about each and about their relationship with each other (as I mention above). There are almost too many models and frameworks for me as I can’t determine which framework is being discussed at which time. The research questions need to be brought together. The ones in the beginning are methodologic and framework research questions which are important, but we never see the research questions of the case study itself, ie., what research questions drove all the data collection processes. Were these even related to the SDG goals and indicators? And if not, how are the SDGs being used? They need to be stated in the beginning of the article and restated in the methods section before presenting the data collection table. Then how do the results of the case study help answer the larger research questions in the beginning. Despite claiming a participatory action research or CBPR research process, I didn’t see evidence that there was participation of the population in decision-making or that there was an ongoing group of citizens that were involved in developing the research questions, deciding the sample, deciding the research steps, collecting the data, interpreting the data, etc. I found this to be an excellent process for collecting data from diverse stakeholders, but I didn’t see the PAR process in action (even though it was declared as a conclusion). Finally, I didn’t see the use of several of the models in the actual case study. For example, how was the SEE methodology used throughout the data collection or interpretation. And where does Table 19 come from? Was it based in any of the methodologies/frameworks mentioned earlier. Again, much is presented, but I’ve lost the connection. 

I do hope this review is helpful and I would greatly value a reworked paper answering these four critiques. The authors clearly have done an extensive research process in one case which is quite impressive, but the leap from data at the case study level, to the larger methodologic interpretation is not quite there.

Author Response

First, we would like to thank the reviewer's comments because these comments have improved the understanding of the article. In this regard, below we specify the changes made in response to these comments.

Point 1. The introduction unfortunately starts off too condensed which ends up being confusing. The first paragraph begins by naming many different frameworks, stating each are important, yet the difficulty for me is in getting a clear picture of what each represents and learning how they are inter-related or build from each other. For example, suddenly there is a list of 17 objectives, 169 goals, (which really should be 17 goals, 169 targets if they are the SDGs) but I have no idea where these are from, or what their intent is (though I assumed they were the Sustainable Development Goals?).  Similarly, participatory action research is named, but there is no definition presented or explanation of why it’s a better approach.  Again, much is implied and condensed, and better connections could be made. For example, there is a call for multiple stakeholders to be involved, which is part of what PAR is, but the authors don’t connect these two ideas as coming from the same orientation.

Response 1: The introduction has been reviewed and the information has been restructured interconnecting the three theoretical references of this article: 2030 Agenda; Strategic Planning and Strategic Environmental Assessment from the participatory action research (lines 29-88)

A much clearer and more inclusive vision is presented with this rewriting of the introduction.

Point 2. I continue to be confused in the framework section, but then some paragraphs begin to offer explanation or a place to orient the reader ie, the paragraphs below:

“From this logic, we can consider municipal management as a typology of activity for municipal organizations whose priority activity is to define goals aimed at improving the quality of life and welfare of citizens in the territory they live, from approaches based on participation and democracy.”

“Currently, the 2030A and the Sustainable Development Goals (SDG) are the frameworks of reference guiding municipal administrative institutions, ensuring that their management model is a sustainable and incorporates sustained strategic planning.”   

Response 2: The framework has been restructured at a general level in response to the comments and contributions of the reviewer to clarify concepts, expand the bibliography and structure the discourse (pages 3-9).

We are aware of the availability of scientific evidence related to the use of PAR as a form of rhetoric to legitimize political decisions. In our case, this investigation arises as a demand of the context agreed by the different agents involved, not only for the political interest and as a consequence, the diversity of agents involved and the ideas expressed.

Even so, a paragraph has been incorporated indicating the idea expressed by the reviewer. (lines 186-191)

Several references of literature on have been added. They analyze the advantages of bottom-up participation models over top-down. The analysis is specified, in our research, in examples of special attention to sectors such as childhood, women and third age. We highlight the role of groups of politicians opposed to municipal government and the value of the youth of many of the representatives (lines 213-224).

Point 3. Each of these could be expanded to introduce this article to provide a better starting point. The research questions and aims provide a better grounding to the article than the first paragraphs do. Yet, these research questions presented above don’t relate sufficiently to the research questions of the case study? In fact, I never get a listing of the research questions for the case study.

Response 3: The objectives and the research questions of the case study have been incorporated according to the general objectives of the research (lines 89-93).

A new table (Table 1) has been included. This table clearly describes four key research ideas that which allow visualizing the relationships between the general approach of the strategic planning model and its application to the case study developed:

1) aims of research, 2) research questions, 3) aims of case study and 4) research questions of case study.

Point 4. Figure one is very helpful. I would also say Figure two is helpful, but there is no explanation of what high and under political leadership is. So to make this figure useful, please provide a good definition of each. 

Response 4: The definition of high political leadership has been added to Figure 2.

Point 5. Figure three is also useful, yet I would value more discussion of why consensus decision-making is favored. I’m not sure I understand how this would be realized at a municipal level, nor is it clear who are the people involved in consensus decision-making.. (policy-makers plus citizen activists? Elected policy-makers? Managers?) Again a lot is assumed without enough explanation.

Response 5: After figure 3 a paragraph has been added explaining how the participatory decision-making process defined in that figure is generated in municipal management.

It also details how this objective is also a reference in the research presented (lines 213-224).

Point 6. Presentation of the SEE is an essential component of this article, but again there is insufficient information for me to grasp how it can be used, or even how it was developed in the first place. The different components are not parallel, as some are content areas and some are processes, which I found confusing.   

Response 6. Information regarding SEE has been expanded.

A new figure (figure 4) has been incorporated that exposes the process to be followed in the SEE and validates, at a conceptual and procedural level, the model followed in the research (lines 228-255).

Point 7 & Point 8. I would value a more straightforward approach, demonstrating how the SEE was applied, instead of another strategic planning model, figure 5.

There is a promise of elaboration and critique of the “model,” but I’m not sure at this point which model is being referred to: Suitability of the information collection instruments used and of their quality. Strengths of the model. Weaknesses of the model. Contribution of the model for the development of sustainable local management. Feasibility and transfer possibilities to other contexts.

Responses 7 & 8. The phases of the SEE have been included in figure 6 to clear the model applied in the research through the SEE process.

This information clarifies that what is intended to be analysed is the model applied in the research (figure 6) that consistently follows the phases of the SEE (360-375).

Point 10. The data collection gets more concrete and I appreciate the specificity of who is involved, etc. And especially love the role of the “letter to the municipality” from the children, but I remain unclear what are the questions being asked, and how do these various data collection strategies relate to the SEE. 

Response 10. The questions related to the questionnaire and the monitoring commission have been incorporated as well as an image about the "letter to municipal representatives" that the children wrote.

A paragraph on how these instruments favour strategic environmental evaluation and citizen participation processes has also been incorporated (Table 2 and Lines 390-399).

Point 11. In the results section introduction, we learn then that an indicator is being used as one of the questions for evaluation, but why didn’t I know this earlier? And where did this indicator come from?  Here is the paragraph, but does this indicator relate to a core research question: “Next, we broadly present the most relevant results achieved after an analysis of the information collected through the different instruments used in the diagnostic phase for the indicator `Citizen perception of environmental and social issues.”

Response 11. In the Methodological framework section, the model defended in the research and the indicators system that emerges from this model are explained.

From it, in this paper one of the indicators is chosen to explain the model: `Citizen perception of environmental and social issues´. For this reason, we address the results in relation to this indicator (lines 360-375).

Point 12. I’m glad to see the factor analysis of the citizen questionnaire, but the factors should be named when the % of variance is presented. Right now I’m left to wonder what the content is until I get to the table.

Response 12. We have divided the table related to factor analysis into two: firs, table 11: Λ total - % variance; and them, table 12: Extraction method: Analysis of main components. Rotation method: Varimax normalization with Kaiser. 

In this way, first the % of variance appears and then the analysis of the factors.

Point 13. On the whole, this article presents several participatory methodologies that I find to be intriguing built on democratic participatory values for municipal planning with homage to various international documents and frameworks, but I find the whole presentation confusing.  Part of the challenge is the translation to English from Spanish, and the different ways the two languages present arguments. Part of the challenge however is that the story is not presented in a straightforward way. It’s a valuable story and I would encourage the authors to make revisions.

Response 13. Timely revisions have been made regarding the language, improving some sentences that were ambiguously written. On the other hand, in general, changes have been made in the body of discourse to eliminate its ambiguity and clarify the models defended and applied in this investigation (check all the sentences written in red that have been the modifications made for this purpose).

Point 14. I have several major critiques. Use of models and frameworks. Although many are mentioned, with images shown, there is insufficient information about each and about their relationship with each other (as I mention above). There are almost too many models and frameworks for me as I can’t determine which framework is being discussed at which time. The research questions need to be brought together. The ones in the beginning are methodologic and framework research questions which are important, but we never see the research questions of the case study itself, ie., what research questions drove all the data collection processes. Were these even related to the SDG goals and indicators? And if not, how are the SDGs being used? They need to be stated in the beginning of the article and restated in the methods section before presenting the data collection table. Then how do the results of the case study help answer the larger research questions in the beginning. Despite claiming a participatory action research or CBPR research process, I didn’t see evidence that there was participation of the population in decision-making or that there was an ongoing group of citizens that were involved in developing the research questions, deciding the sample, deciding the research steps, collecting the data, interpreting the data, etc. I found this to be an excellent process for collecting data from diverse stakeholders, but I didn’t see the PAR process in action (even though it was declared as a conclusion). Finally, I didn’t see the use of several of the models in the actual case study. For example, how was the SEE methodology used throughout the data collection or interpretation. And where does Table 19 come from? Was it based in any of the methodologies/frameworks mentioned earlier. Again, much is presented, but I’ve lost the connection. 

Response 14. We deeply appreciate the effort made by the evaluator by providing four orientations of great importance for the reorientation, reorganization and reflective review of the entire manuscript. We believe that the decisions made accelerate the reading and understanding of the research developed. Likewise, the possibilities of transferring the processes, instruments and methodology have gained after this last revision.

1) The frameworks and the figures associated to the different models have been reviewed providing sufficient information to clarify their interrelation and the logic followed in this article (pages 1-9).

2) The objectives and the research questions of the case study have been incorporated according to the general objectives of the research (Table 1).

3) In the section `Discussion´ different paragraphs have been incorporated of how the study has favoured the PAR (lines 563-570; 606-623).

4) Information on the background of the SEE has been incorporated in the “Methodological Framework” section. The SEE application process and the parallelism with the model applied in the study have been incorporated (lines 229-240 and figure 4 and 6).

5) The conclusions have been restructured by eliminating Table 19 that caused ambiguity. The section has been addressed, exposing the contributions that research has made to the theoretical and methodological field of SEE and ESP (lines 636-648).

Round 2

Reviewer 1 Report

The authors have responded to all of the review comments, and the paper is worth publishing. I only have a few very minor comments:

Page 11, lines 322-330: This text needs to be fixed up, at the moment it has forward slashes and ellipses and looks incomplete. It also needs proper punctuation, with a colon and semi-colons.

Page 13, lines 357-360: The new added text looks incomplete, and some of it is in Spanish.

Page 13, line 371: “I Local Health Plan” – is this correct?

Pages 14 and 18: “councilors” should be “councillors”

Pages 23-25: The new text on these pages mainly consists of lists of issues, but these are not properly punctuated – they need to be introduced with a colon and separated by semi-colons.

Page 25, Conclusion: There is a sentence that doesn’t really make sense (“Arise from this process…”). And the text that has been added probably belongs at the end of the conclusion section rather than the end (you may need to add some linking sentences to make the conclusion flow better).

Author Response

First, we would like to thank the reviewer's comments because these comments have improved the understanding of the article. In this regard, below we specify the changes made in response to these comments (all changes are made in blue within the document).

Point 1. Page 11, lines 322-330: This text needs to be fixed up, at the moment it has forward slashes and ellipses and looks incomplete. It also needs proper punctuation, with a colon and semi-colons.

Response 1. We have improved this text. We have added proper punctuation, with a colon and semi-colons (lines 322-331).

Point 2. Page 13, lines 357-360: The new added text looks incomplete, and some of it is in Spanish.

Response 2. We have reviewed this part of the text and made different modifications to improve understanding. We have also translated the part of the text that was written in Spanish (lines 358-362).

Point 3. Page 13, line 371: “I Local Health Plan” – is this correct?

Response 3. We have change “I Local Health Plan” by “First Local Health Plan”

Point 4. Pages 14 and 18: “councilors” should be “councillors”

Response 4. We change “councilors” by “councillors” (pages 14 and 18).

Point 5. Pages 23-25: The new text on these pages mainly consists of lists of issues, but these are not properly punctuated – they need to be introduced with a colon and separated by semi-colons.

Response 5. We have changed the format of this part of text. We have introduced the list of issues with a colon and separated by full-stop (lines 503-546).

Point 6. Page 25, Conclusion: There is a sentence that doesn’t really make sense (“Arise from this process…”). And the text that has been added probably belongs at the end of the conclusion section rather than the end (you may need to add some linking sentences to make the conclusion flow better).

Response 6. We have reviewed this part of the text and made different modifications to improve understanding (lines 572-583)
